# The Effect of Childhood Psychological Abuse on Depressive Symptoms in Adolescents Exposed to Campus Suicide: The Chain Mediating Role of Psychological Trauma and Anxiety Symptoms

**DOI:** 10.3390/bs15111595

**Published:** 2025-11-20

**Authors:** Tingting Tan, Jiawei Zhao, Mengxuan Wu, Xinyue Zhang, Xinchun Liu, Lili Zhang, Jie Wu

**Affiliations:** 1Key Research Base of Humanities and Social Sciences of the Ministry of Education, Academy of Psychology and Behavior, Tianjin Normal University, Tianjin 300387, China; 2Faculty of Psychology, Tianjin Normal University, Tianjin 300387, China; 3Mental Health Education Center, Tianjin University, Tianjin 300072, China; 4State Key Laboratory of Cognitive Neuroscience and Learning, IDG/McGovern Institute for Brain Research, Beijing Normal University, Beijing 100875, China; 5Department of Communication Sciences and Disorders, The University of Texas Rio Grande Valley, Edinburg, TX 78539, USA; 6Tianjin No. 55 High School, Tianjin 300070, China; 7Tianjin Key Laboratory of Student Mental Health and Intelligence Assessment, Tianjin 300387, China

**Keywords:** childhood psychological abuse, depressive symptoms, anxiety symptoms, psychological trauma, adolescents, suicide exposure

## Abstract

Exposure to campus suicide poses a significant threat to adolescent mental health. While childhood psychological abuse (CPA) is a known vulnerability factor for depression, the mechanisms linking this early adversity to depressive symptoms (DS) following acute trauma remain unclear. This study aimed to test a chain mediation model where CPA contributes to DS through the sequential effects of psychological trauma (PT) and anxiety symptoms (AS). In a cross-sectional study of 1603 adolescents exposed to a campus suicide event, participants completed self-report measures for CPA, PT, AS, and DS. Chain mediation analysis revealed a significant direct effect of CPA on DS. More importantly, the hypothesized chain mediation pathway (CPA → PT → AS → DS) was significant and was identified as the most substantial indirect route. A key asymmetry emerged: the direct effect of CPA on DS remained robust, whereas its direct effect on AS became non-significant when controlling for DS. These findings suggest that CPA establishes a specific vulnerability to depression that, when activated by an acute stressor, initiates a pathological cascade. Interventions for suicide-exposed youth should be trauma-informed, prioritizing those with a CPA history and targeting emergent anxiety to interrupt the progression to severe depression.

## 1. Introduction

Adolescent suicide has become a global public health crisis ([42]). The World Health Organization reports that suicide is the fourth leading cause of death among young people aged 15–29. In the campus environment, witnessing or learning of a peer’s suicide can have profound psychological effects on adolescents, a phenomenon known as “suicide contagion” that is particularly pronounced in this demographic ([22]). Research indicates that adolescents exposed to peer suicide are at a significantly increased risk of developing post-traumatic stress disorder (PTSD), depressive symptoms (DS), and anxiety symptoms (AS) ([1]; [10]; [34]). However, not all adolescents exposed to such traumatic events develop the same degree of psychopathology, which suggests a need to investigate their individual vulnerability factors and the underlying mechanisms.

### 1.1. Childhood Psychological Abuse and Adolescent Mental Health

Childhood psychological abuse (CPA) refers to a sustained pattern of emotionally abusive behaviors by a caregiver, typically manifesting as verbal degradation, threats, neglect, and emotional control ([13]). Compared to physical and sexual abuse, psychological abuse is often more insidious, yet its impact on an individual’s mental health is equally profound ([37]). Research has consistently demonstrated a stable and strong positive association between CPA and adolescent depression and anxiety ([28]; [43]). For example, a large-scale study of over 9000 Chinese adolescents found that CPA was the strongest predictor of both outcomes ([43]). Childhood abuse is known to increase the risk of mental illness in adulthood ([14]; [15]; [26]). A meta-analysis by [32] ([32]) on the relationship between different types of abuse and depression found that CPA is a stronger predictor of depression than childhood physical abuse. Specifically, compared to non-abused individuals, those who experienced emotional abuse and neglect in childhood had a 3.06-fold and 2.11-fold increased risk of developing depression, respectively. Concurrently, individuals who experienced CPA also face a significantly higher risk of developing anxiety disorders ([26]). Importantly, the same study also found that the “type” of abuse is a better predictor of adverse outcomes than the mere “quantity.” Among all types of maltreatment, CPA was the strongest predictor of adolescent anxiety and depression, with explanatory power far exceeding that of other abuse types ([43]). This finding provides a solid empirical basis for the present study’s focus on CPA rather than on mixed or cumulative abuse risks.

### 1.2. The Impact of Psychological Trauma

In the specific context of campus suicide, the relationship between CPA and adolescent mental health becomes more complex. Witnessing or hearing about a peer’s suicide is a highly traumatic experience that often inflicts severe psychological trauma (PT) on survivors ([1]; [10]). According to PTSD theory, a traumatic event triggers intrusive memories, avoidant behaviors, and hypervigilance, negatively impacting an individual’s mental health ([41]). Studies have shown that individuals exposed to another’s suicide are more likely to experience PT and psychological problems ([4]; [34]). Among populations subjected to other types of trauma, [36] ([36]) found that PT mediates the relationship between early abuse and later depression. However, whether this mediating pathway holds true in the unique context of campus suicide exposure requires empirical investigation.

The stress-vulnerability model provides a theoretical framework for understanding the relationship between CPA and post-traumatic psychological responses ([44]). This model posits that mental health problems arise from the interaction between stressors and an individual’s vulnerability. Early life adversities such as CPA, a chronic, trait-level vulnerability, may increase an individual’s sensitivity to stress. This makes them more susceptible to developing state-level psychological problems, such as acute PT, when faced with potent stressors like a campus suicide ([21]). Neurobiological research supports this view, finding that traumatic experiences can lead to persistent dysregulation of the hypothalamic–pituitary–adrenal (HPA) axis, rendering individuals more sensitive to stressors ([18]). Moreover, CPA can reshape the brain’s threat and reward systems, leading to hypersensitivity to threats, which can trigger AS, and blunted responses to pleasure, which can induce DS ([39]). Therefore, it is highly probable that PT plays an important mediating role between CPA and mental health outcomes.

### 1.3. The Directionality of Anxiety and Depression

Anxiety and depression often co-occur and are referred to as the “internalizing spectrum” ([24]). The tripartite model proposes that anxiety and depression share common pathological factors as well as distinct components (e.g., physiological hyperarousal in anxiety, anhedonia in depression) ([6]; [9]). However, the direction of causality between anxiety and depression remains a subject of ongoing debate in academia ([8]; [30]). A prevalent view is that AS typically precede the onset of DS. This perspective is supported by some longitudinal studies, which have found that baseline anxiety levels can predict the severity of subsequent DS ([2]; [5]). Other studies have found that DS can be a precursor to AS, or even that a bidirectional causal relationship exists between them ([2]; [8]; [30]).

In the context of trauma, [20] ([20]), using longitudinal data from the Netherlands study of depression and anxiety (NESDA), showed that CPA is associated with an increased risk for the first onset and recurrence of DS or comorbid disorders, with a history of anxiety disorders mediating this relationship. However, the NESDA did not accurately assess the degree of PT, thus precluding further validation of a chain mediation effect of PT and AS. Nevertheless, the cascade model of developmental psychopathology suggests that early risk factors (like CPA) can trigger a series of maladaptive responses, forming a “chain of problem behaviors” that ultimately leads to long-term psychological disorders ([29]). This chain mediation pathway, “CPA → PT → AS → DS,” is theoretically plausible and aligns with broader frameworks such as the diathesis-stress model and theories on how early trauma impairs emotion regulation, but it has not yet been systematically tested in empirical research.

In summary, although previous studies have separately explored the dyadic relationships among CPA, PT, AS, and DS, the following research gaps remain: (1) few studies have focused on the long-term impact of CPA on adolescent depression within the specific traumatic context of campus suicide exposure; (2) there is a lack of comprehensive examination of the chain mediating roles of PT and AS in the relationship between CPA and DS; and (3) the stability of this mechanism across different gender groups of adolescents needs to be explored. Therefore, based on the cascade model and related theories, this study aims to utilize data collected from a campus crisis intervention to test a series of hypotheses. Specifically, we hypothesized that: (1) CPA would be positively associated with DS, PT, and AS; (2) PT would mediate the relationship between CPA and DS; and (3) a chain mediation pathway would exist, such that CPA would influence DS sequentially through PT and then AS (CPA → PT → AS → DS). By comparing this model with an alternative (CPA → PT → DS → AS), we aim to elucidate the complex underlying mechanism through which CPA contributes to DS in adolescents following a peer suicide event. The findings are expected to offer new perspectives on the interaction between early adversity and current trauma and to provide more targeted empirical evidence for campus-based psychological crisis interventions.

## 2. Method

### 2.1. Participants and Procedure

This study was conducted in the context of a psychological crisis intervention following a student suicide at a middle school. The assessment was conducted within one week of the incident to capture acute psychological responses. As part of the school-wide crisis response launched in the immediate aftermath of the event, all students who had directly witnessed or were otherwise exposed to the incident were invited to participate in a psychological assessment. The term “exposed” included students who directly witnessed the event, those who were in the immediate vicinity, and those who were close friends or classmates of the deceased, as identified by the school’s crisis response team. Due to the nature of the emergency intervention, we did not further stratify the analysis by the type of exposure. In collaboration with school counselors and teachers, informed assent was obtained from all participating students, and written informed consent was secured from their legal guardians.

Volunteering students completed the electronic assessments, which were administered in the school’s computer lab during supervised group sessions. To ensure ethical standards were maintained, school counselors were present during all assessment sessions, and students were explicitly informed that their participation was voluntary, confidential, and would have no bearing on their academic standing. Contact information for psychological support services was provided to all participants.

The final sample consisted of 1603 adolescents, including 806 males (50.28%) and 797 females (49.72%). A majority of participants were an only child (N = 1100; 68.62%), while 503 (31.38%) had siblings. The mean age of the sample was 14.37 years (SD = 1.51, range = 12–17).

### 2.2. Measures

#### 2.2.1. Childhood Psychological Abuse

CPA was assessed using the Child Psychological Maltreatment Scale (CPMS; [33]). This 23-item self-report measure assesses five dimensions of maltreatment: intimidation, neglect, depreciation, interference, and indulgence. Items are rated on a 5-point Likert scale, where higher total scores reflect a greater degree of CPA. In the current study, the scale demonstrated excellent internal consistency (Cronbach’s α = 0.907). A confirmatory factor analysis (CFA) indicated a good model fit to the data: χ^2^/df = 3.676, CFI = 0.943, TLI = 0.934, RMSEA = 0.041, SRMR = 0.040.

#### 2.2.2. Psychological Trauma

Symptoms of PT were evaluated with the Impact of Event Scale-Revised (IES-R; [41]). The validated Chinese version of the IES-R was used in this study. The IES-R is a 22-item questionnaire that measures posttraumatic stress symptoms across three subscales: intrusion, avoidance, and hyperarousal. Participants rate items on a 5-point Likert scale, with higher scores denoting greater symptom severity. A total score of 23 or higher is often used as a clinical cutoff for probable PTSD. In this study, the scale showed high internal consistency (α = 0.933). The CFA results suggested an acceptable model fit: χ^2^/df = 7.191, CFI = 0.902, TLI = 0.890, RMSEA = 0.062, SRMR = 0.054.

#### 2.2.3. Anxiety Symptoms

The severity of AS was measured using the 7-item Generalized Anxiety Disorder scale (GAD-7; [38]). The validated Chinese version of the GAD-7 was used in this study. This widely used screening tool assesses symptom frequency over the past two weeks. Items are rated on a 4-point Likert scale (from 0 = “Not at all” to 3 = “Nearly every day”), with higher total scores representing greater anxiety severity. For the current sample, the GAD-7 demonstrated excellent internal consistency (α = 0.903). The single-factor model showed an excellent fit in the CFA: χ^2^/df = 3.171, CFI = 0.989, TLI = 0.984, RMSEA = 0.037, SRMR = 0.017.

#### 2.2.4. Depressive Symptoms

DS were assessed with the Patient Health Questionnaire-9 (PHQ-9; [23]). The validated Chinese version of the PHQ-9 was used in this study. This 9-item instrument gauges the frequency of depressive symptoms over the preceding two weeks. Each item is rated on a 4-point Likert scale (from 0 = “Not at all” to 3 = “Nearly every day”), with higher total scores indicating more severe depression. The PHQ-9 demonstrated good internal consistency in our sample (α = 0.868). A CFA of the single-factor structure yielded an acceptable to good model fit: χ^2^/df = 7.435, CFI = 0.945, TLI = 0.926, RMSEA = 0.063, SRMR = 0.035.

### 2.3. Data Analyses

Data were analyzed using IBM SPSS 25.0 (IBM Corp., Armonk, NY, USA), with the PROCESS macro (v4.2; [17]) for mediation analyses and Mplus 8.0 (Muthén & Muthén, Los Angeles, CA, USA) for CFAs ([31]). After conducting preliminary correlation and *t*-tests, we tested the proposed simple and chain mediation models. We report both unstandardized (*b*) and standardized (β) coefficients. Indirect effects were tested with 5000 bootstrap resamples and their 95% confidence intervals. Stratified analyses were also performed to examine the model in different subgroups. A two-tailed alpha of 0.05 was used for all significance tests.

## 3. Results

### 3.1. Preliminary Analysis

Prior to the main analyses, we conducted preliminary checks for common method bias and multicollinearity. First, common method bias was assessed using Harman’s one-way test across the four scales (CPMS, IES-R, GAD-7, and PHQ-9). The results showed that the first unrotated factor explained 31.50% of the variance, which is below the 40% threshold, indicating that common method bias was not a significant concern ([35]). Additionally, to check for multicollinearity among the predictor variables in our mediation models, we calculated the Variance Inflation Factor (VIF) and tolerance values. All VIF values were below 2 and all tolerance values were above 0.5, suggesting that multicollinearity was also not an issue.

Table 1 presents the results of the correlation analysis. CPA was significantly and positively correlated with PT, AS, and DS in adolescents (*r* = 0.508, *p* < 0.001; *r* = 0.482, *p* < 0.001; *r* = 0.582, *p* < 0.001, respectively). Following exposure to negative campus events, adolescents’ PT was also significantly and positively correlated with their AS (*r* = 0.667, *p* < 0.001) and DS (*r* = 0.688, *p* < 0.001). AS and DS demonstrated a significant and strong correlation (*r* = 0.770, *p* < 0.001). Additionally, adolescents’ age was significantly and positively correlated with their PT (*r* = 0.050, *p* < 0.05) and DS (*r* = 0.090, *p* < 0.001).

Difference analyses (see Appendix A) revealed that female adolescents reported significantly higher levels of AS (*t* = −1.975, *p* < 0.05) and DS (*t* = −2.738, *p* < 0.01) compared to male adolescents. Furthermore, no significant differences were found between only-child and non-only-child groups across the four main variables (*p*_s_ > 0.05). Consequently, age and gender were included as control variables in all subsequent models (Models A1–A3 and B1–B3). Given that preliminary analyses revealed significant gender differences in AS and DS, and the literature consistently points to gender-specific patterns in the prevalence and expression of internalizing disorders, we opted for stratified analyses. This approach allows for a more nuanced exploration of whether the structural pathways of the mediation models differ between males and females, rather than merely testing for an interaction effect on a single path. In the stratified analyses, age was included as a control variable in all models (Models A2M/F-A3M/F and B2M/F-B3M/F).

### 3.2. Simple Mediation Analysis

To examine the mediating role of PT in the relationship between CPA and mental health outcomes, we constructed mediation models with DS (Models A1–A2) and AS (Models B1–B2) as outcome variables, respectively. Detailed results and descriptions are provided in Appendix A.

Model A1 examined the mediating role of PT in the relationship between CPA and DS (Table 2). The results indicated that CPA significantly predicted PT (β = 0.509, *p* < 0.001), PT significantly predicted DS (β = 0.521, *p* < 0.001, indicating a large effect), and the direct effect of CPA on DS remained significant (β = 0.318, *p* < 0.001). In Model A2, we included AS as a control variable (Table 2). The results showed that even after controlling for AS, the indirect effect of CPA on DS through PT remained significant (CPA → PT: β = 0.245, *p* < 0.001; PT → DS: β = 0.237, *p* < 0.001), with the direct effect maintaining significance (β = 0.221, *p* < 0.001). Bootstrap analysis results also indicated that the 95% confidence interval for the indirect effect did not include zero, confirming the significant mediating effect of PT (Appendix A).

Model B1 examined the mediating role of PT in the relationship between CPA and AS (Table 3). The results demonstrated that CPA significantly predicted PT (β = 0.509, *p* < 0.001), PT significantly predicted AS (β = 0.567, *p* < 0.001), and the direct effect of CPA on AS was significant (β = 0.194, *p* < 0.001). However, when DS was included as a control variable in Model B2 (Table 3), the direct effect of CPA on AS was no longer significant (β = 0.006, *p* > 0.05), although the predictive effect of PT on AS remained significant (β = 0.259, *p* < 0.001). Similarly, bootstrap analysis results indicated that the 95% confidence interval for the direct effect included zero (Appendix A). This finding suggests that when controlling for DS, CPA influences AS primarily through indirect pathways (as shown in Appendix A).

### 3.3. Chain Mediation Analysis

To further investigate the chain mediating roles of PT and AS/DS in the relationship between CPA and DS/AS, we constructed chain mediation models (Model A3 and Model B3). Detailed results are presented in Appendix A.

Model A3 analysis results (Table 2) showed that CPA significantly predicted PT (β = 0.509, *p* < 0.001), PT significantly predicted AS (β = 0.567, *p* < 0.001), and AS significantly predicted DS (β = 0.501, *p* < 0.001, also a large effect). Meanwhile, the direct effect of CPA on DS remained significant (β = 0.221, *p* < 0.001). Bootstrap analysis results indicated that all three indirect effect pathways were significant (Appendix A). Specifically: CPA → PT → DS (indirect effect 1 = 0.043, 95% CI [0.035, 0.052]); CPA → AS → DS (indirect effect 2 = 0.035, 95% CI [0.025, 0.045]); CPA → PT → AS → DS (indirect effect 3 = 0.052, 95% CI [0.044, 0.060]). The total indirect effect was 0.129, accounting for 62.15% of the total effect (Table 4).

In contrast, Model B3 analysis results (Table 3) revealed that while all indirect pathways were significant, the direct effect of CPA on AS was not significant (β = 0.006, *p* > 0.05). This indicates that CPA influences AS entirely through the mediating effects of PT and DS, with the total indirect effect accounting for 98.71% of the total effect (Table 4). A comparative diagram of the chain mediation pathways for Models A3 and B3 is shown in Figure 1.

### 3.4. Stratified Analysis

To verify the robustness of the above findings, we conducted stratified analyses by gender. The results demonstrated that the main findings remained consistent across both male and female adolescent groups. Detailed results and descriptions are provided in Appendix A.

Among male adolescents, Model A2M (Appendix A) showed that after controlling for AS, the direct effect of CPA on DS remained significant (β = 0.248, *p* < 0.001). Chain mediation analysis of Model A3M (Appendix A) indicated that CPA significantly influenced DS through the chain mediating effects of PT and AS (indirect effect 3 = 0.044, 95% CI [0.034, 0.056]), with the direct effect maintaining significance (β = 0.248, *p* < 0.001). However, results from Models B2M and B3M (Appendix A) showed that the direct effect of CPA on AS was not significant (β = 0.007, *p* > 0.05). Bootstrap analysis results (Appendix A) also indicated that the 95% confidence interval for the direct effect included zero. This suggests that among male adolescents, CPA influences AS primarily through indirect pathways.

The analysis results for female adolescents were largely consistent with those for male adolescents. Model A2F (Appendix A) showed that the direct effect of CPA on DS was significant (β = 0.192, *p* < 0.001). Chain mediation analysis of Model A3F (Appendix A) demonstrated that all main variable path coefficients were significant, with the direct effect remaining significant. Similarly to the male adolescent group, results from Models B2F and B3F (Appendix A) showed that the direct effect of CPA on AS was not significant (β = 0.008, *p* > 0.05). Comparative diagrams of the eight model pathways in the stratified analysis are shown in Appendix A.

## 4. Discussion

By constructing and comparing different models within a sample of adolescents exposed to a campus suicide event, this study systematically investigated the complex relationships among CPA, PT, AS, and DS. The results clearly demonstrate that CPA not only directly affects adolescent DS but also significantly exacerbates depression levels through three indirect pathways, most notably via the chain mediation of PT and AS. This finding provides crucial empirical evidence for understanding how early adversity influences adolescent mental health through a cascade of reactions following an acute stressor.

### 4.1. The Impact of CPA on Adolescent PT, AS, and DS

Our findings indicate that CPA is significantly and positively correlated with adolescent PT, AS, and DS, which is consistent with a large body of previous research ([4]; [26]; [32]; [36]; [43]). Individuals who frequently experience degradation and neglect during childhood tend to develop negative self-cognitions and attributional styles. For instance, they attribute negative events to personal deficits and positive events to external factors, leading to feelings of helplessness and psychological problems ([11]; [12]). Moreover, a neuroimaging study by [40] ([40]) found that individuals with a history of CPA exhibit significantly reduced volume in the medial prefrontal cortex, a region closely associated with emotion regulation ([27]). Individuals with CPA experience tend to employ ineffective emotion regulation strategies, such as avoidance and suppression, which often exacerbates the accumulation of negative emotions ([19]). Therefore, in addition to cognitive factors, CPA also impairs individuals’ emotion regulation capacity, thereby increasing the risk of anxiety and depression in adolescents.

Our study further demonstrates that the impact of CPA on adolescent mental health persists when facing a peer’s suicide. For adolescents who have experienced CPA, their cognitive and emotional systems for coping with stress are more fragile, forming a foundation of vulnerability for subsequent traumatic events ([39]). When exposed to a sudden negative life event like a peer suicide, these adolescents with a history of early trauma are more likely to develop intense PT reactions.

### 4.2. The Mediating Role of PT

One of the important findings of this study is the mediating role of PT in the relationship between CPA and adolescent mental health outcomes, validating the applicability of the stress-vulnerability model in the context of campus suicide. Specifically, in the simple mediation analysis, PT explained a substantial portion of the effect of CPA on both DS (45.50%) and AS (59.78%) (Appendix A, Models A1 and B1). It is noteworthy that even after controlling for AS, the predictive effect of CPA on DS remained significant (Model A2). In contrast, after controlling for DS, the direct effect of CPA on AS became non-significant (Model B2). This asymmetry offers important clues about the developmental sequence of anxiety and depression, suggesting that the impact of CPA on DS is specific. CPA not only directly increases depression risk but also indirectly leads to DS by exacerbating individuals’ responses to PT.

This cumulative stress effect can be understood from a neurobiological perspective. Chronic childhood stress (like CPA) can lead to persistent dysregulation of the HPA axis, increasing sensitivity to subsequent stressors ([18]). When an acute traumatic event (like a peer suicide) occurs, the already compromised stress-response system may be overwhelmed, intensifying the severity of both PT and DS ([16]). This suggests that in the aftermath of sudden events like campus suicides, we should promptly identify and focus on high-risk individuals with a history of CPA and provide them with trauma-informed psychological crisis intervention to mitigate their cumulative PT and prevent the onset of severe DS ([7]).

### 4.3. The Chain Mediating Role of PT and AS

The core finding of this study is the validation of the significant “CPA → PT → AS → DS” chain mediation pathway (Model A3), with the direct effect also remaining significant. This pathway accounted for the largest proportion of the total indirect effect (24.82%), revealing a critical developmental pathway from a distal risk factor to the final outcome of DS. Stratified analysis showed that this relational pattern was consistent in both male (Model A3M) and female (Model A3F) adolescents, which enhances the robustness of the findings. This result strongly supports the cascade model in developmental psychopathology, which posits that maladaptation can spread from one domain to another, creating a snowball effect ([29]). Specifically, the trauma vulnerability induced by CPA first manifests as PT symptoms (e.g., flashbacks, hypervigilance) upon encountering a campus suicide event; these persistent PT reactions, particularly hyperarousal symptoms, highly overlap with the core features of generalized anxiety, thereby triggering AS; in turn, a prolonged state of anxiety, accompanied by worry and fear, depletes an individual’s psychological resources, leading to learned helplessness and anhedonia, and ultimately evolving into DS. This finding highlights AS as a critical “bridge” or “waystation” in the progression from traumatic response to DS in post-trauma interventions.

However, when AS was the outcome variable (Model B3), the direct effect of CPA became non-significant, with its influence being almost entirely mediated by PT and DS. Stratified analysis further revealed that this asymmetrical result was consistent across genders (Models B3M and B3F). These findings suggest that although anxiety and depression are highly correlated, their developmental pathways from early risk factors exhibit subtle yet important differences. As a chronic, relational trauma, CPA directly shapes an individual’s core negative beliefs about self-worth and the world (e.g., “I am worthless,” “the world is dangerous”), which are core cognitive features of depression. This aligns with cognitive theories suggesting that early maladaptive schemas are a direct precursor to depressive disorders ([3]). In contrast, the impact of CPA on AS is more dependent on situational triggers (i.e., PT) and comorbidity with DS ([8]). From a symptom perspective, our findings are similar to the network analysis results of [25] ([25]), who found that in adolescents, negative self-esteem (a core symptom of depression) and emotional abuse were the most central nodes in the entire symptom network. The chain mediation model in our study can be understood as a temporal interpretation of such a symptom network, revealing the causal transmission paths between symptoms.

### 4.4. Clinical Implications and Policy Directions

Our findings offer a clear roadmap for school-based crisis interventions. First, in the aftermath of a traumatic event like a campus suicide, universal screening should not only assess for acute trauma but also for histories of early adversity, particularly CPA. This allows for the immediate identification of a high-vulnerability group. Second, a tiered support system is warranted. For all exposed students, psychoeducation on common trauma reactions is essential. For the identified high-risk group (with a CPA history), proactive, trauma-informed counseling should be offered to process both the current event and its resonance with past experiences. Third, our model highlights AS as a critical intervention target. Schools should implement evidence-based techniques for anxiety management (e.g., mindfulness, cognitive restructuring) to act as a “firewall,” preventing the progression from PT to full-blown DS. These steps can transform a reactive crisis response into a proactive, evidence-based mental health support system.

### 4.5. Limitations and Future Directions

Although this study provides important insights into the relationships among CPA, PT, AS, and DS, some limitations exist. First, this study used a cross-sectional design. Although the chain mediation model is theoretically directional, strict causal relationships cannot be inferred from the data alone. Future longitudinal studies should track samples at multiple time points (e.g., 6-month and 1-year follow-ups) to more definitively clarify the temporal sequence and causal links among the variables. Second, all data were self-reported and may be subject to recall bias or the influence of current mood states. Future research could incorporate multi-source data, such as parent/teacher reports or clinical interviews, for cross-validation. Third, our analysis did not control for potential confounding variables such as socioeconomic status or family structure, which could influence both CPA and mental health outcomes. Furthermore, we did not explore potential cultural moderators, such as varying perceptions of abuse or differences in emotional expression, which could influence the observed pathways. Finally, this study focused on risk pathways; future research should also investigate protective factors, such as social support, family cohesion, and adaptive coping skills, which may buffer the negative effects of CPA and trauma. The sample was limited to a single school, so the generalizability of the findings should be approached with caution. Future research also needs to explore the relationships among CPA, PT, AS, and DS in the context of other types of traumatic events to test the cross-situational stability of our findings.

## 5. Conclusions

In the context of peer suicide exposure, this study is the first to systematically validate the chain mediation mechanism through which CPA affects DS via PT and AS. The findings reveal that CPA not only directly influences DS but, more importantly, exerts indirect effects by increasing trauma vulnerability and triggering an anxiety–depression cascade. When providing psychological interventions for adolescents exposed to a campus suicide event, it is crucial not only to address their current trauma symptoms but also to pay special attention to high-risk individuals with a history of CPA. Interventions should not be limited to reducing PT symptoms; they must also closely monitor and promptly address subsequent AS to interrupt the cascade process toward DS, thereby effectively preventing the onset of severe depression. Future intervention protocols should consider integrating trauma-focused therapy, anxiety management techniques, and cognitive–behavioral therapy to form a multi-stage, multi-target integrated intervention framework.

## Figures and Tables

**Figure 1 behavsci-15-01595-f001:**
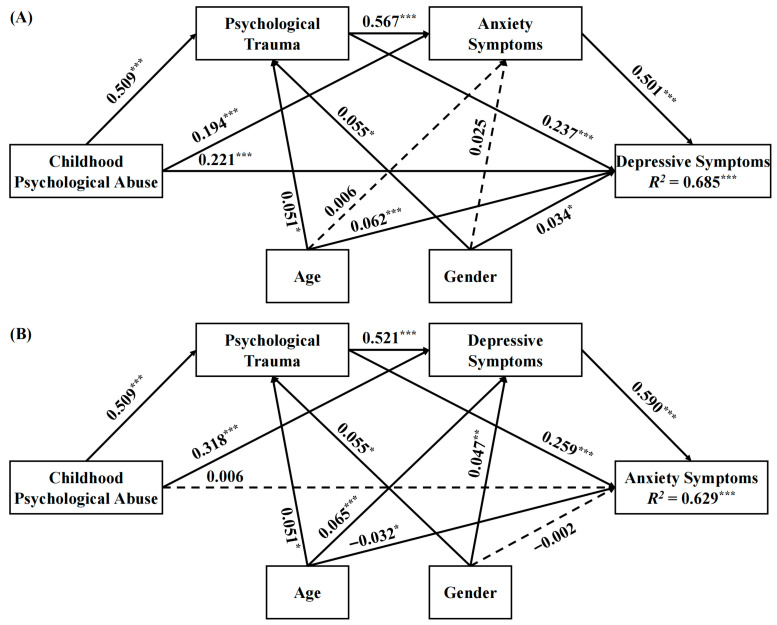
Path diagrams for the chain mediation models A3 and B3 in the full sample. Note: (**A**): Model A3, (**B**): Model B3. Standardized coefficients were presented. * *p* < 0.05, ** *p* < 0.01, *** *p* < 0.001 (two-tailed). Solid arrows indicate statistically significant paths (*p* < 0.05), while dashed arrows indicate non-significant paths.

**Table 1 behavsci-15-01595-t001:** Descriptive statistics, correlations, and Cronbach’s *α* values for variables.

Variable	Mean ± SD	Age	CPA	PT	AS	DS
Age	14.37 ± 1.51	—				
CPA	18.14 ± 13.93	−0.006	—			
PT	29.36 ± 17.95	0.050 *	0.508 ***	—		
AS	4.17 ± 4.69	0.034	0.482 ***	0.667 ***	—	
DS	5.77 ± 4.96	0.090 ***	0.582 ***	0.688 ***	0.770 ***	—
Cronbach’s α			0.907	0.933	0.903	0.868

Note. CPA: Childhood Psychological Abuse; PT: Psychological Trauma; AS: Anxiety Symptoms; DS: Depressive Symptoms. * *p* < 0.05, *** *p* < 0.001 (two-tailed).

**Table 2 behavsci-15-01595-t002:** Model summary information for models A1, A2, and A3.

Model	Path	*b*	*SE*	β	*t*	95% CI (LL/UL)
Model A1	CPA → PT	0.656	0.028	0.509	23.732 ***	0.602/0.710
	CPA → DS	0.113	0.007	0.318	16.358 ***	0.100/0.127
	PT → DS	0.144	0.005	0.521	26.747 ***	0.134/0.155
Model A2	CPA → PT	0.316	0.026	0.245	12.034 ***	0.264/0.367
	CPA → DS	0.079	0.006	0.221	13.185 ***	0.067/0.090
	PT → DS	0.066	0.005	0.237	12.040 ***	0.055/0.076
Model A3	CPA → PT	0.656	0.028	0.509	23.732 ***	0.602/0.710
	CPA → AS	0.065	0.007	0.194	9.195 ***	0.051/0.079
	PT → AS	0.148	0.006	0.567	26.804 ***	0.137/0.159
	CPA → DS	0.079	0.006	0.221	13.185 ***	0.067/0.090
	PT → DS	0.066	0.005	0.237	12.040 ***	0.055/0.076
	AS → DS	0.530	0.021	0.501	25.903 ***	0.490/0.570

Note. Model A1: Simple mediation model, controlling for age and gender; Model A2: Simple mediation model, controlling for anxiety symptoms, age, and gender; Model A3: Chain mediation model, controlling for age and gender. *b*: Unstandardized coefficient; β: Standardized coefficient. *** *p* < 0.001 (two-tailed).

**Table 3 behavsci-15-01595-t003:** Model summary information for models B1, B2, and B3.

Model	Path	*b*	*SE*	β	*t*	95% CI (LL/UL)
Model B1	CPA → PT	0.656	0.028	0.509	23.732 ***	0.602/0.710
	CPA → AS	0.065	0.007	0.194	9.195 ***	0.051/0.079
	PT → AS	0.148	0.006	0.567	26.804 ***	0.137/0.159
Model B2	CPA → PT	0.210	0.028	0.163	7.398 ***	0.154/0.266
	CPA → AS	0.002	0.006	0.006	0.329	−0.011/0.015
	PT → AS	0.068	0.006	0.259	12.135 ***	0.057/0.079
Model B3	CPA → PT	0.656	0.028	0.509	23.732 ***	0.602/0.710
	CPA → DS	0.113	0.007	0.318	16.358 ***	0.100/0.127
	PT → DS	0.144	0.005	0.521	26.747 ***	0.134/0.155
	CPA → AS	0.002	0.006	0.006	0.329	−0.011/0.015
	PT → AS	0.068	0.006	0.259	12.135 ***	0.057/0.079
	DS → AS	0.558	0.022	0.590	25.903 ***	0.516/0.600

Note. Model B1: Simple mediation model, controlling for age and gender; Model B2: Simple mediation model, controlling for depressive symptoms, age, and gender; Model B3: Chain mediation model, controlling for age and gender. *b*: Unstandardized coefficient; β: Standardized coefficient. *** *p* < 0.001 (two-tailed).

**Table 4 behavsci-15-01595-t004:** Total, direct, and indirect effects for models A3 and B3.

Model	Variable	Effect	LLCI	ULCI	Ratio
Model A3	Total effect	0.208	0.194	0.222	—
	Direct effect: CPA → DS	0.079	0.067	0.090	37.85%
	Total indirect effect	0.129	0.116	0.143	62.15%
	Ind 1: CPA → PT → DS	0.043	0.035	0.052	20.68%
	Ind 2: CPA → AS → DS	0.035	0.025	0.045	16.69%
	Ind 3: CPA → PT → AS → DS	0.052	0.044	0.060	24.82%
	Compare 1: Ind 1 − Ind 2	0.008	−0.007	0.023	
	Compare 2: Ind 1 − Ind 3	−0.009	−0.020	0.003	
	Compare 3: Ind 2 − Ind 3	−0.017	−0.030	−0.003	
Model B3	Total effect	0.163	0.148	0.177	—
	Direct effect: CPA → AS	0.002	−0.011	0.015	1.29%
	Total indirect effect	0.161	0.147	0.175	98.71%
	Ind 1: CPA → PT → AS	0.045	0.035	0.055	27.37%
	Ind 2: CPA → DS → AS	0.063	0.052	0.076	38.87%
	Ind 3: CPA → PT → DS → AS	0.053	0.045	0.061	32.47%
	Compare 1: Ind 1 − Ind 2	−0.019	−0.038	0.000	
	Compare 2: Ind 1 − Ind 3	−0.008	−0.023	0.006	
	Compare 3: Ind 2 − Ind 3	0.011	−0.003	0.024	

Note. Ind: Indirect effect. Unstandardized effects were presented.

## Data Availability

The data presented in this study are available on request from the corresponding author. The data are not publicly available due to privacy and ethical restrictions related to the sensitive nature of the event and the vulnerability of the adolescent participants.

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
