# Peer review of "The Effect of Childhood Psychological Abuse on Depressive Symptoms in Adolescents Exposed to Campus Suicide: The Chain Mediating Role of Psychological Trauma and Anxiety Symptoms"

_behavsci, 2025, doi:10.3390/bs15111595_

Round 1

Reviewer 1 Report

Comments and Suggestions for Authors

Reviewer 2 Report

Comments and Suggestions for Authors

The first statement, “Adolescent suicide has become a global public health crisis,” needs a citation.

In the section “Childhood Psychological Abuse and Adolescent Mental Health,” rather than mentioning “numerous studies,” it is recommended to specify which study demonstrates a stable and strong positive association between CPA and adolescent depression and anxiety. Instead of citing at the end of the sentence, begin with the study that establishes this stable and strong positive association between CPA and adolescent depression and anxiety.

In the Methods section, more details are needed about the data collection procedure under the subheading “Participants and Procedure.” It only briefly mentions that “volunteering students completed the electronic assessments.” Although the study obtained IRB approval, it should still describe how data collection procedures maintained ethical standards.

The “Measures” section is well presented.

In Table 1, multiple highly correlated variables were identified. This correlation may raise a red flag regarding the issue of multicollinearity. I wonder whether the authors also tested VIF or tolerance values to ensure there was no multicollinearity problem.

In the Discussion section, it mainly focuses on validating the significant CPA → PT → AS → DS chain mediation pathway. This is fine; however, an important missing point is the discussion of the context in which the data were collected and interpreted, and how these findings are relevant to policy directions. Specifically, the authors should discuss in what context the data were collected and how the results can be interpreted within that framework.

Overall, the paper is well written, and my recommendation is revision.
